

**Secondary Organic Aerosol Formation from Nitrate Radical Oxidation of Styrene: Aerosol**
**Yields, Chemical Composition, and Hydrolysis of Organic Nitrates**
Yuchen Wang[1,2], Xiang Zhang[1], Yuanlong Huang[3], Yutong Liang[2,6], Nga L. Ng*[,2,4,5]
[1] College of Environmental Science and Engineering, Hunan University, Changsha, Hunan, 410082, China
[2] School of Chemical and Bimolecular Engineering, Georgia Institute of Technology, Atlanta, Georgia
30332, USA
[3] Ningbo Institute of Digital Twin, Eastern Institute of Technology, Ningbo, 315200, China
[4] School of Civil and Environmental Engineering, Georgia Institute of Technology, Atlanta, Georgia 30332,
USA
[5] School of Earth and Atmospheric Sciences, Georgia Institute of Technology, Atlanta, Georgia 30332,
USA
[6] Thrust of Sustainable Energy and Environment, The Hong Kong University of Science and Technology
(Guangzhou), Guangdong, 511453, China
*Corresponding Author: Nga L. Ng (ng@chbe.gatech.edu)
**Abstract**
Styrene is emitted from anthropogenic sources and biomass burning and is highly reactive towards
atmospheric oxidants. While it has the highest nitrate radical ($NO_3$) reactivity among aromatic
hydrocarbons, the $NO_3$ oxidation of styrene and formation mechanisms of secondary organic aerosols (SOA)
have not been investigated. In this study, we conduct chamber experiments with styrene concentrations
ranging from 9.5-155.2 ppb. The resulting SOA yields range from 14.0-22.1% with the aerosol mass
loadings of 5.9-147.6 µg/m³ after wall loss corrections. The chemical composition of SOA is characterized
by online measurements, revealing that dimeric organic nitrates (ONs) constitute 90.9% of the total signal
of particle-phase products. $C_{16}H_{16}N_2O_8$ and $C_8H_9NO_4$ are identified as the major particle-phase products,
which constitute 88.3% and 4.1% of the measured signal, respectively. We propose formation mechanisms
for the ON products, including the common $RO_2+RO_2/HO_2$ pathway and other radical chain termination
reactions such as RO+R and R+R. We also investigate the hydrolysis of particulate ONs. The hydrolysis
lifetime for ONs is determined to be less than 30 minutes. This short hydrolysis lifetime can be attributed
to the stabilization of the carbocation by delocalized π orbitals of the benzene-related skeleton of aromatic
ONs. This work provides the first fundamental laboratory data to evaluate SOA production from
styrene+$NO_3$ chemistry. Additionally, the formation mechanisms of aromatic ONs are reported for the first
time, highlighting that compounds previously identified as nitroaromatics in ambient field campaigns could
also be attributed to aromatic ONs.

**Keywords:** Styrene, secondary organic aerosols, aerosol yields, organic nitrate, organonitrate, vapor wall
loss, biomass burning



## 1. Introduction

Aromatic hydrocarbons are a class of unsaturated chemical compounds characterized by the presence of delocalized $\pi$ orbitals. They play a crucial role in the atmosphere, contributing up to 60% of volatile organic compounds (VOCs) in urban environments (Calvert et al., 2002; Cabrera-Perez et al., 2016). Styrene is particularly unique within aromatic hydrocarbons as it possesses both an unsaturated double bond and a benzene ring and has the combined properties of alkenes and aromatic compounds. Although styrene is not the most abundant aromatic hydrocarbon, with concentrations ranging from 0.06 to 45 ppb in the atmosphere (Cho et al., 2014; Tuazon et al., 1993), it has the highest reaction rate constants for reactions with hydroxyl radicals (OH), nitrate radicals ($NO_3$), ozone ($O_3$), and chlorine radicals because of the unsaturated double bond (Tuazon et al., 1993; Tajuelo et al., 2019a, b; Atkinson and Aschmann, 1988; Le Person et al., 2008; Cho et al., 2014). In addition, styrene is the second most efficient aromatic hydrocarbon in forming secondary organic aerosols (SOA) during daytime chemistry, surpassed only by toluene (Sun et al., 2016). SOA yields from OH-initiated photooxidation of styrene can reach as high as around 35% for an aerosol mass loading of 430 $\mu g/m^3$ (Schueneman et al., 2024). A theoretical study suggests that OH-initiated photooxidation of styrene could be a substantial contributor to SOA formation in urban environments (Wang et al., 2015).

To the best of our knowledge, no study has specifically investigated SOA formation from styrene+$NO_3$ oxidation. Although styrene emissions from various anthropogenic sources such as industrial activities, motor vehicle operations, combustion processes, building materials, or consumer products (Zhang et al., 2017; Knighton et al., 2012; Helal and Elshafy, 2012; Okada et al., 2012), are predominantly active during the daytime, high levels of styrene have also been observed at night in urban environments. The nighttime presence of styrene is likely influenced by the boundary layer accumulation effect (Wu et al., 2020; Lu et al., 2023a), which enhances the conditions for styrene+$NO_3$ oxidation. Additionally, biomass burning, particularly wildfires, contributes to emission of styrene in rural and forest regions (Koss et al., 2018). The $NO_3$ oxidation of styrene can occur under conditions such as nighttime chemistry or optically dense plumes during biomass burning events (Decker et al., 2021). Given that styrene exhibits the highest $NO_3$ reactivity among aromatic hydrocarbons (Yang et al., 2020), the styrene+$NO_3$ oxidation can play a major role in the consumption of styrene and formation of SOA.

The $NO_3$ oxidation of VOCs is also expected to generate a substantial quantity of organic nitrates (ONs), primarily through the direct incorporation of the $NO_3$ with the double bond during reactions (Ng et al., 2017; Orel et al., 1978). ONs have been shown to influence $NO_x$ recycling, $O_3$ production, and the formation of SOA in the atmosphere (Ng et al., 2017). Ambient field measurements consistently demonstrate the widespread presence of ONs derived from aromatics in submicron organic aerosols at various locations globally (Lu et al., 2023b; Lin et al., 2021; Jiang et al., 2023; Yang et al., 2019). For



example, in Shanghai, around 16% of the oxygenated organic molecules containing two nitrogen atoms
originate from aromatic compounds (Lu et al., 2023b). In Beijing, the concentration of phenethyl nitrate is
found to be 3.23 ng m$^{-3}$ (Yang et al., 2019). All these suggest that the oxidation of styrene by NO$_3$ could be
an important pathway for generating aromatic ONs.

Hydrolysis of particulate ONs is an important sink of NO$_x$, especially when the ONs have short

hydrolysis lifetimes (Pye et al., 2015; Fisher et al., 2016; Zare et al., 2019; Vasquez et al., 2021; Takeuchi
and Ng, 2019). To our knowledge, there is no study on hydrolysis of ONs formed from oxidation of aromatic
compounds. More studies focus on biogenic ONs. For instance, results from hydrolysis of biogenic ONs in
bulk solutions indicate that the number of alkyl substitutions, the types of functional groups, and the
structures of carbon skeletons are three important factors controlling hydrolysis rates (Darer et al., 2011;
Hu et al., 2011; Jacobs et al., 2014; Rindelaub et al., 2016; Wang et al., 2021b). A common feature identified
in the mechanisms is the formation of stable carbocations, which facilitates the rapid hydrolysis of ONs.
Previous research indicates that the benzene-related skeleton, featuring three delocalized π orbitals,
enhances the hyper-conjugation effect and stabilizes the carbocation (Wang et al., 2021b). Consequently,
ONs produced from styrene+NO$_3$ oxidation, which include such benzene-related skeletons, are likely to
have short hydrolysis lifetimes. However, this hypothesis has not been evaluated before, because of the lack
of hydrolysis studies of aromatic ONs.

In this study, we aim to investigate SOA formation and chemical composition from styrene+NO$_3$

oxidation. We conduct a series of chamber experiments in the dark under both dry and humid conditions.
SOA yields are determined across a wide range of initial styrene concentrations under dry conditions. The
chemical composition of SOA is characterized by online mass spectrometry and the SOA formation
mechanism is proposed based on these measurements. Additionally, we investigate hydrolysis of particulate
ONs. These results can be used to estimate SOA formation and transformation from NO$_3$ oxidation of
styrene from anthropogenic emissions and biomass burning in ambient environments.
**2.  Experimental Section**
**2.1. Environmental chamber experiments**

The experimental conditions are summarized in Table 1. All experiments are performed in the Georgia

Tech Environmental Chamber (GTEC) Facility, which consists of two 12 m$^3$ Teflon chambers (Boyd et al.,
2015). Experiments are conducted at 295 ± 3 K and ambient pressure. Most experiments are conducted
under dry conditions (RH < 3%, Exp. 1-10), with the exception of two experiments (Exp. 11 and 12), which
are conducted under humid conditions (approximately 50% and 70%, respectively). These humid
experiments allow for the investigation of the hydrolysis processes of SOA.

A typical experiment begins with the injection of seed particles into the chamber by atomizing a dilute

ammonium sulfate solution (AS; 0.015 M). Subsequently, styrene (99 %, Sigma-Aldrich) is injected into



the chamber using a glass bulb, where the evaporation of styrene is facilitated by the flow of zero air at a
rate of 5 L min$^{-1}$ through the bulb. The initial particle number and volume concentration are $2.9 \times 10^4$
particles cm$^{-3}$ and $3.2 \times 10^{10}$ nm$^3$ cm$^{-3}$, respectively. The initial concentration of styrene ranges from 9.5-
155.2 ppb. It is noted that Exp. 9 and 10 do not involve seed particle injection and are conducted specifically
to determine the density of styrene+NO$_3$ SOA.
N$_2$O$_5$ is generated by the reaction of NO$_2$ (Matheson, 500 ppm) and O$_3$ (generated by passing purified
air through a UV light (Jelight 610), ~125 ppm) in a flow tube (0.8 L min$^{-1}$ flow rate, 115 s residence time)
and injected into the chamber as NO$_3$ precursor. The injection time ranges from 5 to 75 minutes, depending
on the initiation concentration of styrene. The typical styrene to N$_2$O$_5$ ratio is approximately 1:2. To ensure
that styrene is predominantly oxidized by NO$_3$, the concentrations of O$_3$ and the flow rates of both NO$_2$ and
O$_3$ are adjusted (based on results from a simple kinetic box model) to optimize N$_2$O$_5$ production while
minimizing O$_3$ concentration. Upon entering the chamber, N$_2$O$_5$ thermally decomposes, generating NO$_2$
and NO$_3$, establishing an equilibrium that marks the onset of NO$_3$ oxidation (Boyd et al., 2015; Takeuchi
and Ng, 2019).
**2.2. Gas- and particle-phase measurements**
The concentrations of O$_3$ and NO$_x$ are monitored with an ultraviolet absorption O$_3$ monitor (Teledyne
T400) and a NO$_x$ monitor (Thermo Fisher Scientific 42C) (Teledyne 200EU), respectively. A gas
chromatograph with flame ionization detector (GC-FID, Agilent) is used to track the decay of styrene.
Aerosol volume and size distributions of particles smaller than 1 μm in electrical mobility diameter are
measured by a scanning mobility particle sizer (SMPS) under the low-flow mode (sheath flow of 2 L min$^{-1}$
). The SMPS is consisted with a differential mobility analyzer (TSI 3080) and a condensation particle
counter (TSI 3775).
A high-resolution time-of-flight aerosol mass spectrometer (HR-ToF-AMS; Aerodyne Research Inc.)
is used to quantitatively measure the bulk particle-phase chemical composition including organics, nitrate,
sulfate, ammonium, and chloride. The working principle and operation of the HR-ToF-AMS are described
in detail elsewhere (DeCarlo et al., 2006). Elemental analysis of the data is conducted to determine the
elemental composition of the bulk aerosols (Canagaratna et al., 2015). The data are analyzed using PIKA
v1.16I in Igor Pro 6.38B.
The speciated oxidized gas- and particle-phase products are measured using a high-resolution time-of-
flight chemical-ionization mass spectrometer coupled with the filter inlet for gases and aerosols (FIGAERO-
CIMS; Aerodyne Research Inc.) with iodide (I$^-$) as the reagent ion. Details on the operation of the instrument
has been described in previous literature (Boyd et al., 2017; Takeuchi and Ng, 2019; Nah et al., 2016b;
Chen et al., 2020; Lopez-Hilfiker et al., 2014). Briefly, reagent ions are generated from a cylinder containing
a mixture of CH$_3$I and dry N$_2$ (Airgas) and through polonium-210 source (NRD; model P-2021). The



instrument measures gaseous compounds by sampling air from the chamber at 1.7 L min$^{-1}$. At the same
time, particles in the chamber are collected onto a polytetrafluoroethylene filter with the sampling rate from
1 to 5 L min$^{-1}$ depending on the aerosol mass concentrations. A gradually heated nitrogen gas flows over
the filter, evaporating oxidized organic species and transporting them into the CIMS for detection. The data
are analyzed using Tofware v2.5.11. All the compounds presented in this study are I$^-$ adducts.
**2.3. Volatility Calibration**
In the FIGAERO-CIMS, during the thermal desorption stage, the temperature at which the maximum
($T_{max}$) desorption signal for a particle-phase compound is observed corresponds to effective saturation mass
concentration (C*) (Lopez-Hilfiker et al., 2014; Thornton et al., 2020; Stark et al., 2017; Ylisirniö et al.,
2021). The experimental procedures for volatility calibration have been described in detail in our previous
study (Takeuchi et al., 2022). Briefly, the relationship between $T_{max}$ and C* in FIGAERO-CIMS is
established by depositing a mixture of standards with known C* onto the filter. These standards are then
subjected to thermal desorption using the same thermal program applied in the chamber experiments. The
standards used in this study include glycolic acid ($C_2H_4O_3$), oxalic acid ($C_2H_4O_2$), malonic acid ($C_3H_3O_4$),
succinic acid ($C_4H_6O_4$), meso-erythritol ($C_4H_{10}O_4$), levoglucosan ($C_6H_{10}O_5$), suberic acid ($C_8H_{14}O_4$), azelaic
acid ($C_9H_{16}O_4$), sebacic acid ($C_{10}H_{18}O_4$), dodecanedioic acid ($C_{12}H_{22}O_4$), palmitic acid ($C_{16}H_{32}O_2$), stearic
acid ($C_{18}H_{36}O_2$), and behenic acid ($C_{22}H_{44}O_2$). The relationship between C* (in μg m$^{-3}$) at 25 °C and $T_{max}$
(in °C) obtained in this study is $\log_{10} [C^*_{25°C}] = -0.085T_{max} + 5.12$ (Figure S1) and is consistent with the
calibrations in Takeuchi et al. (2022).
**3. Results**
**3.1. SOA formation from NO$_3$ radical oxidation of styrene**
A series of chamber experiments with different initial styrene concentrations is performed to
investigate SOA formation from NO$_3$ oxidation of styrene (Table 1). In these experiments, the ratio of
styrene to N$_2$O$_5$ is maintained at 1:2 to optimize the reaction conditions, facilitating a complete consumption
of styrene and allowing for the analysis of the resulting products. Figure S2 presents the time series of the
formation of SOA during a typical experiment (Exp. 7). In all experiments, styrene is fully reacted within
60 minutes, and the peak aerosol concentration is typically observed within the same time range (Figure
S2).
All SOA data are corrected for particle wall loss by applying size-dependent coefficients determined
from wall loss experiments (Nah et al., 2017). The nucleation experiments are conducted to determine SOA
density. By comparing SMPS volume distribution and HR-ToF-AMS mass distribution (Bahreini et al.,
2005; Alfarra et al., 2006; Ng et al., 2008), the SOA density is determined to be 1.35 g/cm$^3$. Figure 1 shows
the SOA yields (Y, 4.5%–16.1% for Exp. 1-8, over a wide range of aerosol mass loadings (ΔM$_O$), 1.9–107.4
μg/m$^3$). For all experiments, peak aerosol mass concentration is obtained from the SMPS aerosol volume



concentration (averaged over 30 minutes at peak aerosol loading) and the calculated aerosol density. SOA
yields are parametrized as a function of organic mass produced using the a semi-empirical model (Odum et
al., 1996, 1997) based on gas-to-particle partitioning of two semi-volatile products (Eq.1). The fitting molar
yields ($\alpha_1$ and $\alpha_2$) are 0.1 and 0.09, and the fitted partitioning coefficients ($K_1$ and $K_2$) are 0.4 and 0.02 ($R^2$
= 0.997).
$$Y = \Delta M_O \left[ \frac{\alpha_1 K_1}{1 + K_1 M_O} + \frac{\alpha_2 K_2}{1 + K_2 M_O} \right]$$   Eq. 1
**3.2. Chemical composition of SOA**
A typical HR-ToF-AMS aerosol mass spectrum is shown in Figure 2 along with the National Institute
of Standards and Technology (NIST) mass spectra of possible styrene+$NO_3$ oxidation products. There are
a few notable ions in the aerosol mass spectrum. The signals at $m/z$ 39 ($C_3H_3^+$), 50 ($C_4H_2^+$), 51 ($C_4H_3^+$), 52
($C_4H_4^+$), 77 ($C_6H_5^+$), 78 ($C_6H_6^+$), 91 ($C_7H_7^+$), 105 ($C_7H_5O^+$), and 106 ($C_7H_6O^+$) are aromatic compound
signatures with benzene ring (McLafferty and Turecek, 1993). The signals at $m/z$ 91 ($C_7H_7^+$), 105 ($C_7H_5O^+$),
and 106 ($C_7H_6O^+$), while not particularly significant in the mass spectra of other aromatic SOA systems (Yu
et al., 2014, 2016; Zhang et al., 2023; Liu et al., 2022; Chen et al., 2021), are relatively high for styrene+$NO_3$
oxidation system. However, $m/z$ 91 ($C_7H_7^+$) is a signature ion for SOA formed from $NO_3$ oxidation of β-
pinene (Boyd et al., 2015) and photooxidation of β-caryophyllene (Tasoglou and Pandis, 2015). $m/z$ 91
($C_7H_7^+$) has also been detected as one of the major fragments of synthetic monoterpene ON standards
measured by HR-ToF-AMS (Takeuchi et al., 2024). Therefore, only $m/z$ 105 ($C_7H_5O^+$) and 106 ($C_7H_6O^+$)
can potentially serve as useful indicators for SOA formed from styrene oxidation in ambient aerosol mass
spectra. Note that the HR-ToF-AMS spectrum of styrene+$NO_3$ oxidation is very similar to the NIST mass
spectra of benzaldehyde ($C_7H_6O$) and 2-hydroxy-1-phenyl ethanone ($C_8H_8O_2$). However, we do not detect
the prominent peaks of dimers in the HR-ToF-AMS, which can be explained by instability of dimer under
the high collision energy of the instrument.
Figure S3a presents the time series of organics and nitrate as measured by HR-ToF-AMS from a typical
experiment. Sulfate is used to normalize the decay of organics and nitrate because it is non-volatile and any
decrease in sulfate is reflective of particle wall loss and changes in aerosol collection efficiency (CE) in the
HR-ToF-AMS (Henry and Donahue, 2012). Organics and nitrate exhibit similar decay trends. However, the
situation differs when examining the time series of major organic families relative to sulfate (Figure S3b).
Hydrocarbon fragments ($C_xH_y$ Family), $C_xH_yO$ family, and $C_xH_yO_zN$ (z>1) family exhibit similar decay
rates, but decrease more rapidly relative to sulfate than $C_xH_yON$ and $C_xH_yO_z$ families. This may indicate
that further aerosol aging leads to the formation of more oxidized fragments ($C_xH_yO_z$) (Boyd et al., 2015).
This may also suggest that the aging products are more likely to produce $C_xH_yON$ rather than $C_xH_yO_zN$
fragments.





FIGAERO-CIMS is used to measure speciated particle-phase composition of styrene+NO$_3$ SOA,
including both dimeric and monomeric products. The characteristic SOA mass spectrum from FIGAERO-
CIMS (Figure 3a) is categorized according to molecule types: CHO, CHON, and CHON$_2$, each contains
compounds with different numbers of carbon atoms (Figure 3b). The SOA composition is dominated by
nitrogen-containing compounds, with C$_x$H$_y$O$_z$N$_2$, C$_x$H$_y$O$_z$N, and C$_x$H$_y$O$_z$ molecules constituting 91.8%,
7.4%, and 0.8% of the measured signal, respectively (Figure 3c). Dimers (with carbon numbers C$_9$ to C$_{16}$)
make up 90.9% of the signal, with the majority being C$_{16}$H$_x$O$_z$N$_2$ dimers constituting 89.4% of the total
signal. Figure 4 shows the temporal evolution of major particle-phase products, the dominant product is
C$_{16}$H$_{16}$N$_2$O$_8$, contributing 88.3% of the total signal. The next most abundant particle-phase product is
C$_8$H$_9$NO$_4$, which constitutes 4.1% of the total signal. In addition, C$_8$H$_7$NO$_4$, C$_8$H$_9$NO$_5$, C$_8$H$_8$N$_2$O$_6$, and
C$_8$H$_8$N$_2$O$_7$ are major monomeric particle-phase products. C$_{16}$H$_{14}$N$_2$O$_8$, C$_{16}$H$_{17}$NO$_7$, C$_{15}$H$_{13}$NO$_6$, and
C$_{16}$H$_{13}$NO$_6$ are major dimeric particle-phase products. It is noted that it is possible that the compounds
detected as monomeric species are formed from the thermal decomposition process in the FIGAERO-CIMS
(Yang et al., 2021; Kumar et al., 2023; Stark et al., 2017).

**3.3. Hydrolysis of styrene-derived organic nitrates**

Building on results from our previous study (Wang et al., 2021b), the unique benzene-related skeleton
of styrene ONs can facilitate their rapid hydrolysis. Therefore, we conduct experiments with three different
chamber RH, including dry (RH<3%), RH~50%, and RH~70% to study the hydrolysis of styrene-derived
ONs. Figures S4a illustrate the time series of nitrate measured by HR-ToF-AMS for these different RH
systems. Distinct variations are observed in nitrate levels across different chamber RH conditions. The
presence of the small amounts of nitrate prior to the commencement of experiments under different RH
conditions could potentially result from the uptake of background nitric acid onto aqueous seed particles
(McMurry and Grosjean, 1985; Grosjean, 1985; Matsunaga and Ziemann, 2010; Zhang et al., 2014, 2015;
Yeh and Ziemann, 2015; La et al., 2016; Nah et al., 2016a; Krechmer et al., 2016; Huang et al., 2018). After
N$_2$O$_5$ is injected into the chamber, the large increase in nitrate in the higher RH experiments can be attributed
to the reactive uptake of N$_2$O$_5$ and/or the dissolution of HNO$_3$ into aqueous aerosols (Takeuchi and Ng,
2019), subsequently neutralized by ammonia to form ammonium nitrate. Therefore, to evaluate the extent
of particle phase ONs hydrolysis, the contributions of inorganic nitrate (NO$_{3,Inorg}$) and ONs (NO$_{3,Org}$) to the
measured nitrate from HR-ToF-AMS need to be separated.
We differentiate the contributions of NO$_{3,Inorg}$ and NO$_{3,Org}$ to the measured nitrate based on the method
reported in Farmer et al., (2010), as described in Eq. 2.
$$x = \frac{(R_{obs}-R_{NH_4NO_3})(1+R_{ON})}{(R_{ON}-R_{NH_4NO_3})(1+R_{obs})}$$      Eq. 2
Where R$_{NH4NO3}$ (*i.e.*, NO$^+$/ NO$_2^+$ from ammonium nitrate) is derived from the standard ionization



efficiency (IE) calibration of HR-ToF-AMS using 300 nm-sized ammonium nitrate particles, and the value
is 1.8. The $R_{ON}$ (*i.e.*, $NO^+/NO_2^+$ for ONs) value is dependent on the aerosol chemical composition and
instrument. The $NO^+/NO_2^+$ ratio throughout all the dry experiments remains constant at approximately 4.9,
referred to as $R_{ON}$ in this study. The $R_{ON}/R_{NH4NO3}$ ratio (ratio of ratio, RoR) is 2.7, which falls within the
range of previous studies with the presence of ONs (Day et al., 2022). The $R_{obs}$ value is 2.8 and 3.8 for
experiments with RH ~70% and 50%, corresponding to $NO_{3,Org}$ contributions of 51% and 80%, respectively.
Figures S4b and S5 depict the time series of $NO_{3,Org}$ for experiments under different RH. We also compare
the $NO_{3,Org}$ measured by HR-ToF-AMS and speciated ONs measured by FIGAERO-CIMS, which show
similar trends (Figure S5).
Here, we follow the approach reported in our previous work and used $_pON/OA$ ratio to evaluate the
extent of ON hydrolysis via Eq. 3 (Takeuchi and Ng, 2019; Takeuchi et al., 2024). It is noted that $_pON$ refers
to the total mass concentration of particulate ONs, encompassing both the organic and nitrate components
of the ON compounds. Similarly, OA represents the total mass concentration of organic aerosols, which
includes both nitrated and non-nitrated organic compounds.
$$\frac{_pON}{OA} = \left(\frac{NO_{3,org}}{Organic + NO_{3,org}}\right) \times \left(\frac{MW_{_pON}}{MW_{NO_2,ON}}\right) = \left(\frac{\frac{NO_{3,org}}{Organic}}{1 + \frac{NO_{3,Org}}{Organic}}\right) \times \left(\frac{MW_{_pON}}{MW_{NO_2,ON}}\right) \qquad \text{Eq. 3}$$

Where $MW_{pON}$ refers to the average molecular weight of $_pON$ estimated from FIGAERTO-CIMS data.
Assuming uniform sensitivity among detected species, $MW_{pON}$ is similar across different experiments,
within the range of 182.7-184.0 g mol$^{-1}$. $MW_{NO_2,ON}$ is the molecular weight of the nitrogen-containing
moiety of ONs (*i.e.*, $NO_2$, 46 g mol$^{-1}$) measured by the HR-ToF-AMS, as discussed in detail in a recent
study by Takeuchi et al. (2024).
As illustrated in Figure 5a, the time series of $_pON/OA$ stabilizes fairly quickly, similar to what we have
observed previously for monoterpene systems (Takeuchi and Ng, 2019). This suggests that the hydrolysis
lifetime of ONs that undergo hydrolysis is no more than 30 minutes. The hydrolyzable fraction of styrene-
derived ONs can be estimated from the difference in $_pON/OA$ between dry and RH experiments once the
ratio stabilizes. The hydrolyzable fraction is about 52.7-60.6%. The observed hydrolyzable ONs are
$C_{16}H_{16}N_2O_8$, $C_8H_7NO_4$, and $C_8H_9NO_4$, as determined by comparing the FIGAERO-CIMS mass spectra
under dry and RH 70% conditions (Figure 5b). The non-nitrated organic species (*i.e.*, $C_8H_8O_5$) is enhanced
correspondingly due to the hydrolysis of ONs.
**4. Discussion**
**4.1 SOA yields over a wide range of organic mass loadings**
There is no prior study on SOA formation from styrene+$NO_3$ oxidation, but previous research has
reported SOA formation from photolysis, OH-initiated photooxidation, and ozonolysis of styrene (Figure
S6). Photolysis of styrene results in the lowest SOA yields, ranging from 1.8%-3.6%, in the presence of



29.4 to 202.7 μg/m³ of ΔM$_O$ (Tajuelo et al., 2019b). Different peroxy radical (RO$_2$) chemistry, controlled
by the concentration of NO, influences the SOA formation during OH-initiated photooxidation of styrene.
The SOA yields are observed to be 4.0-5.0% with 174.4-348.3 μg/m³ of ΔM$_O$ in conditions where RO$_2$+NO
chemistry dominated (Tajuelo et al., 2019b), and around 2-35 % with 2.8-430 μg/m³ of ΔM$_O$ where
RO$_2$+RO$_2$ chemistry prevailed (Yu et al., 2022b; Schueneman et al., 2024). Several previous studies have
reported that the SOA yields from the ozonolysis of styrene (Ma et al., 2018; Na et al., 2006) are higher
than those from OH-initiated photooxidation of styrene under RO$_2$+NO chemistry, but lower than those
under RO$_2$+RO$_2$ chemistry. As shown in Figure S6, the SOA yield from the styrene+NO$_3$ oxidation is higher
than in other styrene oxidation systems when ΔM$_O$ is lower than 80 μg/m³.

Organic vapor wall loss has been reported to impact SOA yield calculation and can lead to an

underestimation of SOA yields by as much as a factor of 4 (McMurry and Grosjean, 1985; Grosjean, 1985;
Matsunaga and Ziemann, 2010; Zhang et al., 2014, 2015; Yeh and Ziemann, 2015; La et al., 2016; Nah et
al., 2016a; Krechmer et al., 2016; Huang et al., 2018). Therefore, to evaluate the potential effect of organic
vapor wall loss on SOA yields in our study, experiments without seed particles are carried out (Exp. 9 and
10). As show in Figure S7, the SOA formation from nucleation experiments (without seed particles) is lower
than condensation experiments (with seed particles). Zhang et al. (2014) determine that if organic vapor
wall loss is significant in chamber experiments, the addition of more seed particles will lead to an increase
in SOA yields. The differences in SOA formation between styrene+NO$_3$ nucleation experiments and
condensation experiments suggest the possible impact of organic vapor wall loss. Although particle wall
loss is accounted for, the SOA yields reported in Figure 1 and Table 1 could represent the lower limit owing
to vapor wall loss, where organic vapors could have partitioned to the chamber wall rather than to the
aerosols.

To account for the impact of vapor wall loss, we employ the semi-empirical equation (Eq. 1) for SOA

yield to correct for vapor wall loss. The correction relies on two assumptions: (1) styrene+NO$_3$ oxidation
yields two major products, and (2) the partition of these major products between gas and particle phases, as
well as vapor wall loss is controlled by C* of these products. The detailed procedures and the relevant
equations are shown in Supplementary Information (SI). The SOA yields and the SOA yield curve after this
correction is shown in Figure 1 and Table S1. The fitted molar yields ($\alpha_1$ and $\alpha_2$) are 0.8 and 0.1, and the
fitted partitioning coefficients (K$_1$ and K$_2$) are 8.1×10$^{-4}$ and 7.5 (corresponding to C* values of 1.2×10$^3$
μg/m³ and 1×10$^{-1}$ μg/m³, respectively) after vapor wall loss correction ($R^2$=0.991). The vapor wall loss bias
factor, defined as the ratio of aerosol mass loadings after correction to those before correction, ranges from
1.3 to 3.2. The vapor wall loss bias factor is smaller for the experiments with the higher organic mass
loadings, consistent with results reported by Zhang et al. (2014).

In styrene+NO$_3$ oxidation, two products stand out in abundance, C$_8$H$_9$NO$_4$ and C$_{16}$H$_{16}$N$_2$O$_8$, which





constitute 92.4% of the total particle signal (Figure 3). We utilize two methods to estimate the C* of
$C_8H_9NO_4$ and $C_{16}H_{16}N_2O_8$: the Estimation Program Interface Suite (EPI Suite) method and FIGAERO
thermal desorption method (Takeuchi et al., 2022). The C* of $C_8H_9NO_4$ and $C_{16}H_{16}N_2O_8$, as simulated by
EPI Suite, are $1.6\times10^3$ $\mu g/m^3$ and $9.0\times10^{-1}$ $\mu g/m^3$, respectively. The FIGAERO thermal desorption profiles
for $C_8H_9NO_4$ and $C_{16}H_{16}N_2O_8$ are shown in Figure S8. The $T_{max}$ of $C_8H_9NO_4$ and $C_{16}H_{16}N_2O_8$ are 24.6 and
67.8 °C, respectively, corresponding to C* values of $1.1\times10^3$ and $2\times10^{-1}$ $\mu g/m^3$. The C* of $C_8H_9NO_4$ and
$C_{16}H_{16}N_2O_8$, estimated by these two methods are similar to those obtained from the semi-empirical equation
(Eq. 1). The consistency of the C* values across different methods supports the applicability of the semi-
empirical equation (Eq. 1) proposed in this study for correcting vapor wall loss in SOA yield calculation.
**4.2 Proposed formation mechanisms for styrene-derived organic nitrate monomers and dimers**

We present the proposed formation mechanisms of major particle-phase products detected by

FIGAERTO-CIMS (Figure 6). The formation mechanisms involve two distinct routes to generate the
monomeric and dimeric ONs respectively. In monomeric ONs formation pathway (Pathway A): The
reaction begins with the addition of a nitrooxy group (-$ONO_2$) onto the double bond of styrene follow by
$O_2$ addition to form a nitrooxy peroxy radical (via R1). This addition is expected to be favored because the
$NO_3$ radical adds to the less substituted carbon atom in the double bond. We also propose the formation
mechanism with $NO_3$ radical adduct to the more substituted carbon atom in the double bond in Figure S9.
The nitrooxy peroxy radical can further react with $RO_2$ or hydroperoxyl radical ($HO_2$) to produce the styrene
hydroxy nitrate ($C_8H_9NO_4$), ketone nitrate ($C_8H_7NO_4$), and peroxy nitrate ($C_8H_9NO_5$) via reactions R2, R3,
and R4, respectively. $C_8H_9NO_4$, which is the second most abundant product, has been observed in ambient
environments as well. However, it has been assumed to be nitroaromatic compounds (R-$NO_2$, *i.e.*,
dimethoxynitrobenzene) instead of aromatic ONs (R-$ONO_2$) (Kong et al., 2021; Wang et al., 2021a;
Salvador et al., 2021). $C_8H_9NO_5$ and $C_8H_7NO_4$ are also identified from residential wood-burning boilers
and have been suggested to be nitroaromatic compounds (Salvador et al., 2021) formed from the OH-
initiated photooxidation of phenolic VOCs in the presence of NO (Vione et al., 2001, 2004; Jenkin et al.,
2003; Frka et al., 2016; Vidović et al., 2018). Considering the ambient concentrations of nitroaromatic
compounds remain high even at night (Wang et al., 2019), our work suggests that styrene-derived ONs
could account for these molecular species in ambient environments, often only attributed to nitroaromatic
compounds. Furthermore, the nitrooxy peroxy radical can react with $NO_3$ radical to produce alkoxyl (RO)
radical (via R6). The RO radical can further undergo decomposition to produce benzaldehyde, benzene
hydroxy aldehyde, and $NO_2$ as the byproduct. $NO_2$ can react with nitrooxy peroxy radical as well through
reaction R5 to produce styrene peroxy nitrate ($C_8H_8N_2O_7$). Peroxy nitrate remains unstable at room
temperature unless there is a carbonyl group present to induce an electron-withdrawing effect, thereby
stabilizing the peroxy nitrooxy group (Yu et al., 2022b). Here, the benzene ring can also stabilize the peroxy





nitrooxy group through electron coupling (McMurry, 2012), thus promoting the formation of styrene peroxy
nitrate. According to Lewis and Moodie (1996), NO$_3$ radical can react with the double bond in olefins to
produce the ONs containing two nitrooxy groups. We utilize the same mechanism to elucidate the formation
of C$_8$H$_8$N$_2$O$_6$ in our work (via R7).
The formation of nitrogen-containing dimeric products from aromatic oxidation systems has been
observed in laboratory chamber studies (Molteni et al., 2018; Kumar et al., 2023; Mayorga et al., 2021).
For example, Kumar et al. (2023) report that dimeric products make up 54.2% of the total particle signal in
NO$_3$ oxidation of an aromatic hydrocarbon mixture, with the majority (42.3%) being C$_x$H$_y$O$_z$N dimers.
Nitrated diphenyl ether dimers have been observed from the NO$_3$ oxidation of phenolic VOCs, as reported
by Mayorga et al. (2021). In our work, the predominant dimeric product is C$_{16}$H$_{16}$N$_2$O$_8$, which is generated
from RO$_2$+RO$_2$ reaction (via R8). Additionally, C$_{16}$H$_{14}$N$_2$O$_8$, C$_{16}$H$_{17}$NO$_7$, C$_{15}$H$_{13}$NO$_6$, and C$_{16}$H$_{13}$NO$_6$ are
also major dimeric ONs products observed in the particle phase. Benzaldehyde, ketone nitrate (C$_8$H$_7$NO$_4$),
and benzene hydroxy aldehyde, can further react with NO$_3$ radical via reactions R9, R10, and R11 to
generate radicals A1, A2, and A3, respectively. A2 and A3 can further react with each other (via R12) to
terminate the radical reaction by producing C$_{16}$H$_{13}$NO$_6$. Additionally, A3 can react with RO radical from
reaction R5 to form C$_{16}$H$_{14}$N$_2$O$_8$ (via R13), while A1 can react with nitrooxy peroxy radical to produce
C$_{15}$H$_{13}$NO$_6$ (via R14). We propose the formation of C$_{16}$H$_{17}$NO$_6$ through the reaction between nitrooxy
peroxy radical and styrene (via R15), followed by the RO$_2$ + RO$_2$ reaction (via R16). It is noted that in this
work, we propose formation mechanisms of dimeric compounds based on molecular formulas of the
detected species. Further experimental studies and density functional theory work are necessary to confirm
the structures and formation of these dimeric compounds.
**4.3 Hydrolysis of organic nitrates formed from styrene+NO$_3$ oxidation**
Condensed-phase hydrolysis has been identified as a significant sink for ONs, evidenced by substantial
ON uptake to aerosols and the reported short hydrolysis lifetimes (Pye et al., 2015; Fisher et al., 2016; Zare
et al., 2019; Vasquez et al., 2021). Recent studies have reported experimentally constrained parameters for
the hydrolysis of biogenic ONs derived from monoterpene or isoprene, through chamber or bulk solution
experiments (Takeuchi and Ng, 2019; Morales et al., 2021; Hu et al., 2011; Darer et al., 2011; Jacobs et al.,
2014; Rindelaub et al., 2016; Vasquez et al., 2021; Wang et al., 2021b). Studies using bulk solutions to
exam hydrolysis of ONs with specific structures have demonstrated that the number of alkyl substitutions,
functional groups, and carbon skeletons are three important factors controlling hydrolysis rates (Wang et
al., 2021b). Therefore, the hydrolysis lifetimes of biogenic ONs can be as fast as seconds or minutes, or as
stable as several days without observable hydrolysis, depending on the structures of the ONs. Unlike bulk
solutions, which only involve aqueous solutions, chamber experiments can simulate the hydrolysis of ONs
formed from VOC oxidations in aerosol water. In Takeuchi and Ng (2019), the ON hydrolysis lifetimes are



determined to be less than 30 minutes for NO₃ oxidation and OH-initiated photooxidation of α-pinene and
β-pinene systems. The hydrolysis lifetime of ONs formed from OH-initiated photooxidation of β-ocimene
has been found to be pH-dependent, 51 (±13) minutes at a pH of 4 and 24 (±3) minutes at a pH of 2.5
(Morales et al., 2021).
There is limited study on the hydrolysis of anthropogenic ONs. Only one study reported the hydrolysis
lifetime of ONs resulting from OH-initiated photooxidation of 1,2,4-trimethylbenzene to be 6 hours (Liu et
al., 2012). The extended hydrolysis lifetime of ONs from this system can be explained by the cleavage of
the benzene ring, which disrupts the delocalized π orbitals. In this study, we determine that the hydrolysis
lifetime for ONs from styrene+NO₃ oxidation is no more than 30 minutes. Based on our previous work with
bulk solutions (Wang et al., 2021b), we propose that the benzene-related skeleton of aromatic ONs, which
contains three delocalized π orbitals, can lead to rapid hydrolysis. This is because the delocalized π orbitals
enhance the hyperconjugation effect and stabilize the carbocation, thereby decrease the hydrolysis lifetimes
of ONs (Wang et al., 2021b). This mechanism helps explain the short hydrolysis lifetimes observed in this
work.
The fraction of hydrolyzable ONs is crucial for understanding the role of hydrolysis as a loss
mechanism for ONs and NOₓ. Takeuchi and Ng (2019) reported that the hydrolysable fraction of ONs from
the NO₃ oxidation and OH-initiated photooxidation of α-pinene and β-pinene systems range from 9-36%.
However, more than 50% of the ONs resulting from the OH-initiated photooxidation of 1,2,4-
trimethylbenzene are hydrolyzable (Liu et al., 2012). In our study, we observe that the fraction of
hydrolyzed styrene-derived ONs ranges from 52.7% to 60.6%. Overall, while research on the fraction of
hydrolyzable ONs is still limited, these findings so far indicate that the hydrolyzable fraction of ONs
resulting from the oxidation of aromatic VOCs are larger than those from biogenic VOCs. The difference
can likely be explained by the fact that only ONs with specific chemical structures, particularly tertiary
nitrates in biogenic VOCs oxidation systems, are susceptible to hydrolysis. In contrast, a large fraction of
aromatic ONs features structures with delocalized π orbitals, which facilitate hydrolysis.
**5. Atmospheric implications**
To the best of our knowledge, this study is the first to demonstrate the formation of SOA and particulate
ONs from styrene+NO₃ oxidation. We systematically carry out a series of chamber experiments with initial
styrene concentrations ranging from 9.5 to 155.2 ppb under dry and humid conditions at room temperature.
The resulting SOA yields range from 4.5% to 16.1% with the aerosol mass loadings of 1.9 to 107.4 μg/m³.
It is known that the loss of organic vapors to the chamber wall can lead to underestimation of SOA yields.
For instance, Zhang et al. (2014) compare the results from a statistical oxidation model and experimental
observations and determine that the correction factor for SOA yields to range from 2.1 to 4.2. Here, we use
a semi-empirical model that incorporates the gas-to-particle partitioning of two semi-volatile products to



correct for the effect of vapor wall loss on SOA yields, the correction factor is found to range from 1.3 to
3.2, consistent with previous studies (Zhang et al., 2014). By applying the correction factor derived in this
work, the corrected SOA yields range from 14.0% to 22.1% with the aerosol mass loadings of 5.9 to 147.6
$\mu g/m^3$. The yields obtained in this study provide the basis to determine the contributions of styrene+NO$_3$
chemistry to SOA formation. Styrene has been detected at the ppb levels in ambient atmosphere and has a
high emission factor from biomass burning, with typical abundance ranging from 0.06 to 45 ppb (Cho et
al., 2014; Tuazon et al., 1993; Yu et al., 2019; Koss et al., 2018). These are in the range of our experiments,
corresponding to the formation of up to 32.5 $\mu g/m^3$ of SOA with the vapor wall loss correction. Our results
serve as fundamental inputs for parameterizing SOA formation from styrene in atmospheric models.
We examine the chemical composition of SOA and propose formation mechanisms for the major
monomeric and dimeric ON products detected in the particle phase. We find that dimeric products constitute
90.9% of the signal, while monomeric products account for only 9.1%. Previous studies have identified
nitroaromatic compounds (R-NO$_2$) as the nitrogen-containing products from aromatic VOCs in ambient
conditions (Kong et al., 2021; Wang et al., 2021a; Salvador et al., 2021). Our work introduces an alternative
perspective, suggesting that ONs could also be nitrogen-containing products from the oxidation of aromatic
VOCs in the atmosphere. Dimeric nitrogen-containing compounds from the oxidation of aromatic VOCs
(*e.g.*, toluene, p-xylene, ethylbenzene, 1,3,5-trimethylbenzene, phenol, cresol, 2,6-dimethylphenol, and etc.)
have been observed in chamber experiments (Molteni et al., 2018; Kumar et al., 2023; Mayorga et al., 2021).
Here, based on speciated molecular level characterization of SOA, we are able to propose the chemical
structures and formation mechanisms of dimeric ON products for the first time. Besides the common
RO$_2$+RO$_2$ pathway, we also suggest that other radical chain termination reactions, such as RO+R or R+R,
could explain the formation of the major dimeric ONs. Further density functional theory calculations and
experimental work are needed to provide additional evidence for our proposed mechanisms. In contrast to
chamber experiments, the detection of dimeric nitrogen-containing compounds derived from aromatics is
rare in field campaigns. Ye et al. (2021) observe C$_{\geq 19}$H$_y$O$_z$N$_{1-2}$ compounds in ambient aerosols in Shenzhen,
China by FIGAERO-CIMS, exhibiting a ring and double-bond equivalence (RDBE) exceeding 10 and an
aromaticity equivalent (Xc) surpassing 2.70 (Yassine et al., 2014; Wang et al., 2017). These compounds can
be considered as aromatic ONs (Table S2). Considering the difference between controlled laboratory
experiments and the complexity of ambient environments, it will be intriguing to explore why dimeric ONs
derived from aromatic compounds are rarely observed in the field.
In this study, we observe that the hydrolysis lifetime of styrene-derived ONs (about 52.7 to 60.6% of
total ONs) is no more than 30 minutes. This finding supports our previous assumptions about the
relationship between hydrolysis lifetimes and the molecular structures of ONs (Wang et al., 2021b). The
unique delocalized π orbitals provided by the benzene-related skeleton of styrene ONs can stabilize the



carbocation, thereby promoting hydrolysis. The hydrolysis lifetime observed for ONs generated from
styrene+NO$_3$ oxidation can serve as experimentally constrained parameter for modeling hydrolysis of
aromatic ONs in general. For example, not only styrene-derived ONs but also other aromatic ONs such as
furan or methylfuran ONs (Joo et al., 2019), despite lacking benzene ring, have the potential to undergo
rapid hydrolysis due to the presence of delocalized $\pi$ orbitals. The hydrolysis lifetimes are crucial for
regional and global chemical transport models to accurately assess the impacts of hydrolysis of aromatic
ONs on the nitrogen budget and subsequent ozone formation.

**Data availability.**
The chamber experiment data are available online at the Index of Chamber Atmospheric Research in the
United States (ICARUS, https://icarus.ucdavis.edu/).

**Supporting Information**
The Supporting Information is available free of charge at: https:// /acp.copernicus.org. Additional details
on volatility calibration, HR-ToF-AMS and FIGAERO-CIMS mass spectra of SOA from styrene+NO$_3$
oxidation, representative product distribution, time series data from HR-ToF-AMS for different RH
experiments, literature review of previous styrene oxidation studies, method for correcting vapor wall loss
when determining SOA yields, and other proposed mechanisms for the major particle-phase products.

**Author contributions**
YW and NLN designed the research. YW conducted the experiments. YW, XZ, and NLN interpreted the
data and wrote the paper. YL conducted volatility calibration and YH helped with vapor wall loss correction.
All the authors discussed the results and commented on the paper.

**Competing interests.**
The authors declare that they have no conflict of interest.

**Acknowledgements**
The authors would like to acknowledge financial support by the Young Scientists Fund of the National
Nature Science Foundation of China (Grants 22306059), the National Science Foundation (NSF) CAREER
AGS-1555034 and the National Oceanic and Atmospheric Administration (NOAA) NA18OAR4310112.
This work was also supported by the Science and Technology Innovation Program of Hunan Province
(Grants 2024RC3106), the Science and Technology Planning Project of Hunan Province (Grants



2023JJ40128), the Nature Science Foundation of Changsha (Grant kq2208019), the Fundamental Research
Funds for the Central Universities (Grant 531118010830), and State Key Laboratory of Loess and
Quaternary Geology, Institute of Earth Environment (Grant SKLLQG2235). The FIGAERO-CIMS was
purchased through NSF Major Research Instrumentation (MRI) grant 1428738. The authors would also like
to acknowledge Dr. Long Jia for kindly providing the raw data in his paper (Yu et al., 2022a) for Figure S6.

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



**Table 1**. Summary of experimental conditions in this study.

| Exp | RH | Seed | $\Delta HC$ (ppb)[b] | $\Delta HC$ ($\mu g\ m^{-3}$) | $N_2O_5$ (ppb) | $HC:N_2O_5$ Ratio | $\Delta M_O$ ($\mu g\ m^{-3}$)[c,d] | SOA Mass yield (%) |
|---|---|---|---|---|---|---|---|---|
| 1 | <3% | AS[a] | 9.5 ± 0.3 | 40.7 ± 1.3 | 20 | 1:2 | 1.9 ± 0.2 | 4.5 ± 0.5 |
| 2 | <3% | AS | 15.6 ± 0.2 | 67.1 ± 0.8 | 36 | 1:2 | 5.4 ± 0.2 | 8.1 ± 0.4 |
| 3 | <3% | AS | 18.0 ± 0.1 | 77.6 ± 0.5 | 40 | 1:2 | 6.8 ± 0.3 | 8.6 ± 0.4 |
| 4 | <3% | AS | 28.4 ± 0.5 | 122.0 ± 2.0 | 60 | 1:2 | 12.4 ± 0.6 | 10.1 ± 0.5 |
| 5 | <3% | AS | 48.2 ± 1.0 | 207.2 ± 4.2 | 100 | 1:2 | 26.1 ± 0.4 | 12.6 ± 0.2 |
| 6 | <3% | AS | 80.1 ± 0.7 | 344.4 ± 3.1 | 160 | 1:2 | 48.5 ± 0.7 | 14.1 ± 0.2 |
| 7 | <3% | AS | 99.1 ± 0.8 | 426.0 ± 3.4 | 200 | 1:2 | 67.0 ± 1.0 | 15.7 ± 0.2 |
| 8 | <3% | AS | 155.2 ± 1.8 | 667.5 ± 7.9 | 310 | 1:2 | 107.4 ± 0.5 | 16.1 ± 0.1 |
| 9 | <3% | None | 18.2 ± 0.5 | 78.3 ± 2.2 | 40 | 1:2 | 0.6 ± 0.1 | 0.81 ± 0.1 |
| 10 | <3% | None | 100.4 ± 0.8 | 431.8 ± 3.5 | 200 | 1:2 | 47.6 ± 0.8 | 11.03 ± 0.2 |
| 11 | 50% | AS | 17.3 ± 0.2 | 74.5 ± 0.8 | 40 | 1:2 | /[e] | /[e] |
| 12 | 70% | AS | 15.8 ± 0.5 | 67.8 ± 1.9 | 40 | 1:2 | /[e] | /[e] |

[a.] Ammonium sulfate. [b.] Uncertainties in hydrocarbon concentration are calculated from 1 standard deviation from GC-FID measurements. [c.] Uncertainties in aerosol mass loading are calculated from 1 standard deviation of aerosol volume as measured by the SMPS. [d.] Density is 1.35 g m$^{-3}$, calculated from the comparison of HR-ToF-AMS and SMPS size distribution data. [e.] These numbers are not reported because the density of SOA in humid experiments is not available.




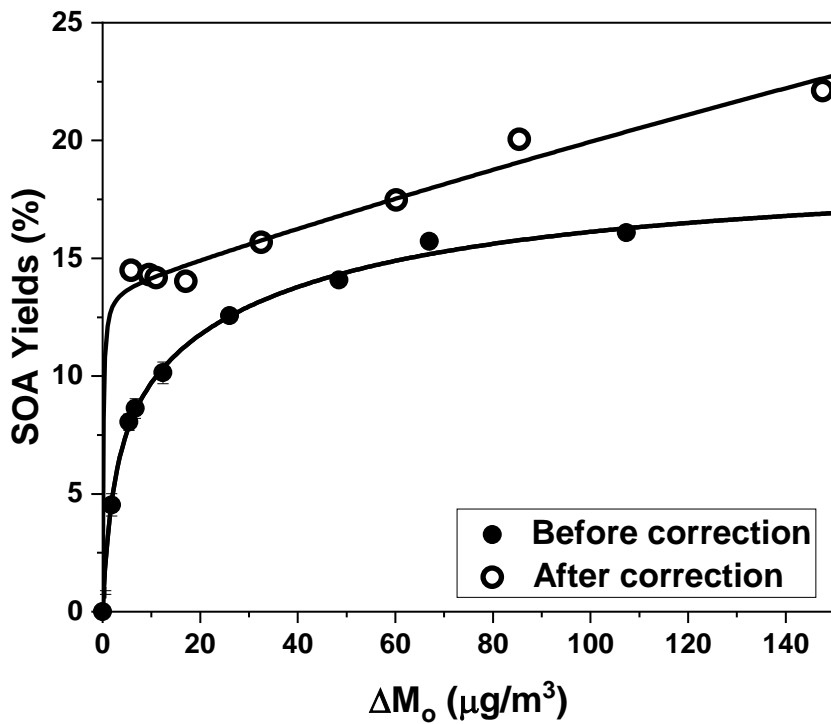


**Figure 1.** SOA yield data and yield curves for styrene+NO$_3$ oxidation with and without vapor wall loss

correction.








**Figure 2**. Comparison between the National Institute of Standards and Technology (NIST) mass spectra of


(a) benzaldehyde; (b) 2-hydroxy-1-phenyl ethenone; and (c) the HR-ToF-AMS mass spectrum (in integer
*m/z*) of SOA from styrene+NO₃ oxidation. The chemical structures for benzaldehyde and 2-hydroxy-1-
phenyl ethenone are shown in (a) and (b).

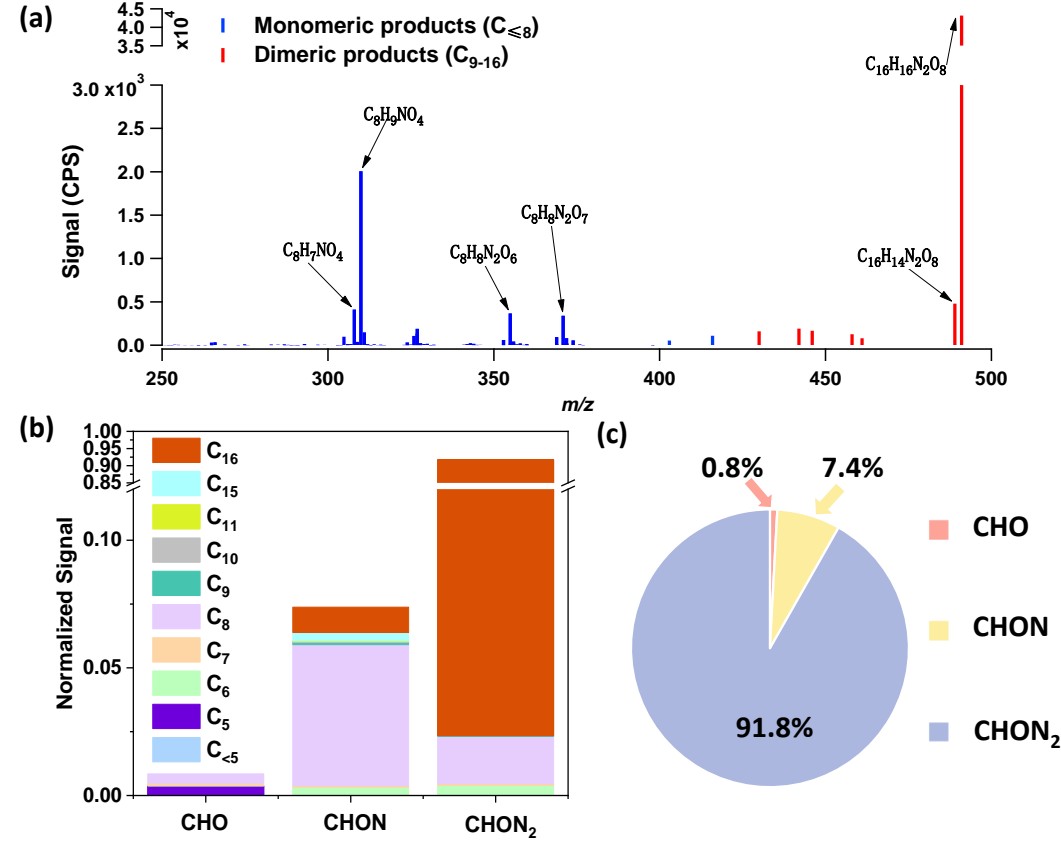

**Figure 3**. (a) A typical FIGAERO-CIMS mass spectrum of SOA from styrene+$NO_3$ oxidation (Exp. 7). The sticks are colored to distinguish between monomeric and dimeric products, as indicated in the legend. Prominent masses are labeled with the corresponding chemical formulae without an iodide ion. Only *m/z* 250-500 are shown here, there are no specific major products before *m/z* 250 except $I^-$ and $HNO_3I^-$ ; (b) SOA product distribution, categorized by molecule types: CHO, CHON, and $CHON_2$. Each category is further subdivided by carbon number; (c) The pie chart illustrates the relative percentage contributions of CHO, CHON, and $CHON_2$.





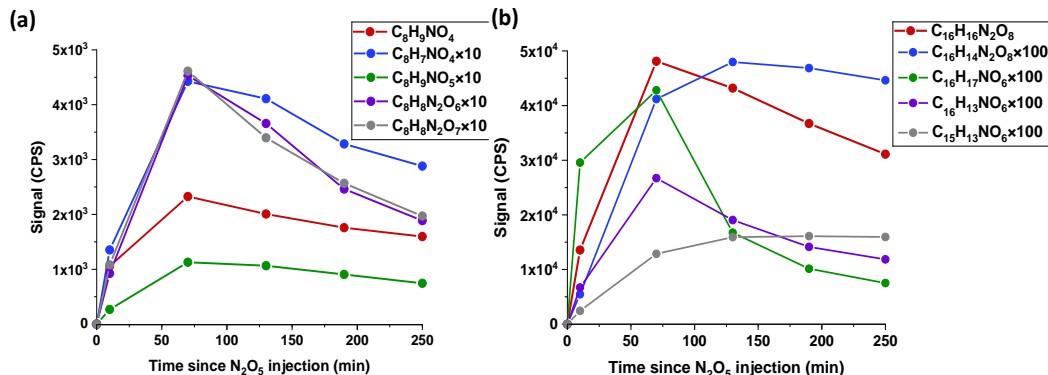

**Figure 4.** The time series of major particle-phase products from styrene+NO₃ oxidation measured by
FIGAERO-CIMS, including: (a) monomeric styrene ONs; and (b) dimeric styrene ONs.



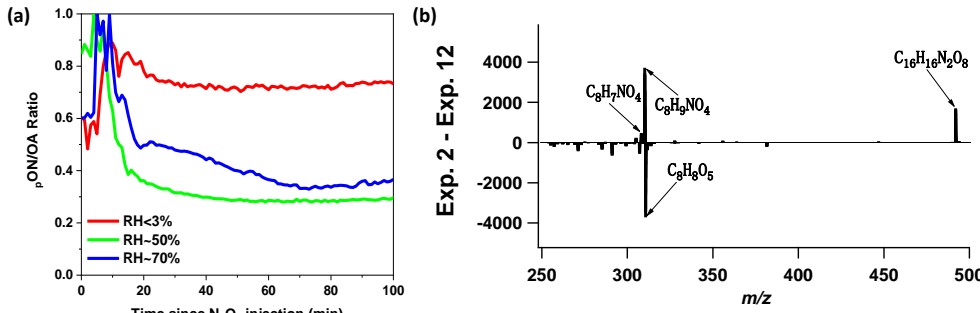


**Figure 5. (a)** Time series data of the ratio of particulate ONs ($_p$ON) to total organic aerosols (OA) in
Exp. 2 (RH<3%), Exp. 11 (RH~50%), and Exp. 12 (RH~70%); (b) FIGAERO-CIMS difference mass
spectrum of SOA: Exp. 2 (RH<3%) minus Exp. 12 (RH~70%). Prominent masses are labeled with the
corresponding chemical formulae without an iodide ion.




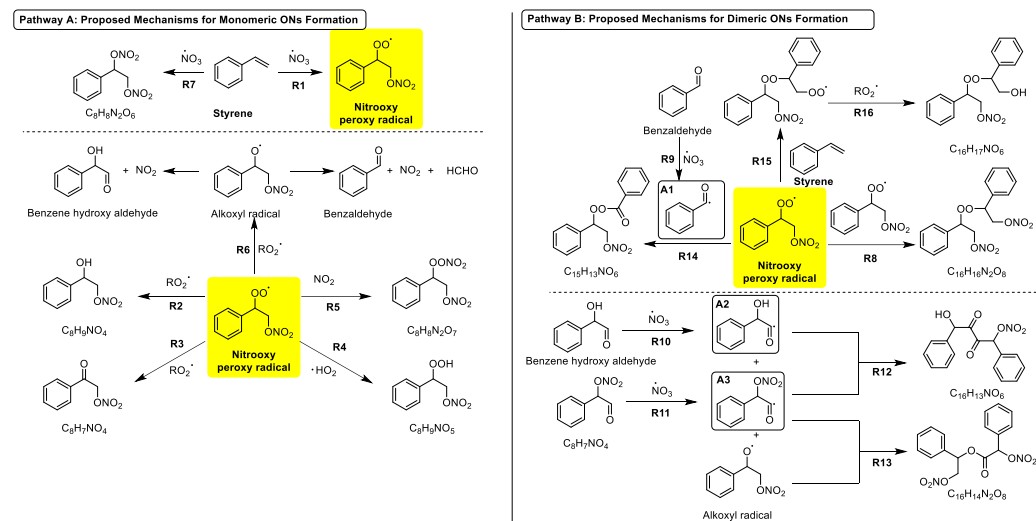

**Figure 6.** The proposed formation mechanisms for the major particle-phase products resulting from
styrene+NO$_3$ oxidation includes two distinct pathways: Pathway A is the proposed formation pathway for
monomeric ON products. Pathway B is the proposed formation pathway for dimeric ON products. Radicals
A1, A2, and A3 are highlighted in the boxes as the major products resulting from the reaction between
aldehydes and NO$_3$ radicals. The nitrooxy peroxy radical is highlighted in yellow as the major RO$_2$ in
mechanisms.