# Peer review of "Secondary Organic Aerosol Formation from Nitrate Radical Oxidation of Styrene: Aerosol Yields, Chemical Composition, and Hydrolysis of Organic Nitrates"

_EGUsphere, 2024_

## Author Comment (AC1)

**Point-by-point response to reviewer comments**

Manuscript ID: egusphere-2024-3849
Title: "Secondary Organic Aerosol Formation from Nitrate Radical Oxidation of Styrene: Aerosol Yields, Chemical Composition, and Hydrolysis of Organic Nitrates"
Author(s): Wang, Yuchen; Xiang Zhang; Yuanlong Huang; Yutong Liang; Nga L. Ng

We thank the reviewers for their thoughtful comments. Each of the comments has been addressed and detailed in our point-by-point response below. The exact comment from the reviewers is in black and italic style while our response is in blue and normal format in this document. The revised text as it appears in the revised manuscript is in green. The figure appears in this response (labelled R1) that is not identical to figure in the manuscript and supporting information section and is meant to clarify our response.

**Reviewer: 1**

**General comments**

*This manuscript demonstrates the formation of SOA and particulate organic nitrates (ONs) from styrene + NO3 oxidation. The SOA yields of styrene were 4.5% to 16.1% with the aerosol mass loading of 1.9 to 107.4 ug m-3. The chemical composition of SOA was detected by aerosol mass spectrometer (AMS) and a high-resolution time-of flight chemical-ionization mass spectrometer coupled with the filter inlet for gases and aerosol (FIGAERO-CIMS). It is interesting that the C16H16N2O8 and C8H9NO4 were identified as the major particle-phase products, which constitute 88.3% and 4.1% of the measured signal. The main mechanisms for those formation products were RO2+RO2/HO2, RO+R and R+R. In addition, the hydrolysis of ONs was not more than 30 minutes.*

*This work is suitable to publish in Atmosphere Chemistry and Physics. However, before publishing in ACP, this manuscript needs major revisions to address my comments:*

**Response**:

We appreciate the viewpoints of the reviewer. Each of the comments has been addressed and detailed in our point-by-point response below.

**Comment**

*1. Line 24 and 52: Please change the ug/m³ into ug m⁻³ through the manuscript.*

**Response**

Revised as suggested.

*2. Line 37-38. There are no keywords in ACP journal.*

**Response**

Thanks for pointing this out, we have deleted the keywords.

*3. Line 116-118. This manuscript has mentioned that they used a simple kinetic box model to optimize N2O5 production while minimizing O3 concentration. And I did not see the results on how you used the model optimize the N2O5 production. In addition, I am curious whether O3 was also injected into the chamber during the experiment. If yes, what is the concentration?*

**Response**

(1) Thanks for pointing this out. We have added the following to the revised SI to clarify this section and in lines 114-116 in main text: "To ensure that styrene is predominantly oxidized by $NO_3$, the concentrations of $O_3$ and the flow rates of both $NO_2$ and $O_3$ are adjusted (based on results from a simple kinetic box model, Table S1) to optimize $N_2O_5$ production while minimizing $O_3$ concentration."

SI:

**Model Calculation for $N_2O_5$ concentration**

A simple chemical model is used to adjust the concentration of $O_3$ and flow rates of both $NO_2$ and $O_3$ in flow tube to maximize the production of $N_2O_5$, such that styrene is dominantly oxidized by $NO_3$ radical. This simple chemical model is developed using the Master Chemical Mechanism (MCM, version 3.3.1). The related reactions and their rate constants are shown in following.

**Table S1**. List of reactions and their rate constants for estimation of $N_2O_5$ in flow tube.

| Reaction | Rate Constant |
|----------|---------------|
| $NO_2 + O_3 \rightarrow NO_3 + O_2$ | $3.5 \times 10^{-17}$ cm$^3$ molecules$^{-1}$ s$^{-1}$ |
| $NO_2 + NO_3 \rightarrow N_2O_5$ | $6.7 \times 10^{-12}$ cm$^3$ molecules$^{-1}$ s$^{-1}$ |
| $N_2O_5 \rightarrow NO_2 + NO_3$ | $2.2 \times 10^{-1}$ s$^{-1}$ |

(2) We do not inject $O_3$ directly into chamber; instead, we use a flow tube to generate $N_2O_5$. As we mention in lines 110-113: "$N_2O_5$ is generated by the reaction of $NO_2$ (Matheson, 500 ppm) and $O_3$ (generated by passing purified air through a UV light (Jelight 610), ~125 ppm) in a flow tube (0.8 L min-1 flow rate, 115 s residence time) and injected into the chamber as $NO_3$ precursor, similar to our prior studies (Boyd et al., 2017; Takeuchi and Ng, 2019; Takeuchi et al., 2022)."

*4. Line 129-134. Did you calibrate size distribution and ionization efficiency of aerosol mass spectrometer in this manuscript? If yes, provide the value. What is collection efficiency of AMS do you use in this manuscript?*

**Response**

Yes, we conducted ionization efficiency (IE) calibration and size calibration. The IE calibration was performed with 300 nm ammonium nitrate particles during the styrene+$NO_3$ experiments. The value is provided in lines 241-243: "Where $R_{NH4NO3}$ (*i.e.*, $NO^+ / NO_2^+$ from ammonium nitrate) is derived from the standard ionization efficiency (IE) calibration of HR-ToF-AMS using 300 nm-sized ammonium nitrate particles, and the value is 1.8." We did not calculate collection efficiency (CE) as we did not need to use this value in our analysis.

*5. The FIGAERO-CIMS heating program is not clear. E.g., how long did you do ramp and desorption, etc.? Please describe the heating procedure in manuscript. Line 142-143, the author used different sampling 1 to 5 L min-1 depending on the aerosol concentration. How long did you collect the aerosol*

*on filters? In addition, Figure 4 and S5 show the concentration of major compounds from CIMS. However, the units of compounds are counts per seconds (cps). Those values are strongly dependent on the aerosol mass in the filters. It is better to use the ppt or ng m-3 units to remove the aerosol loading effect.*

**Response**

(1) We have added more details to the FIGAERO-CIMS program description to clarify. In lines 139-145: "Each sampling cycle lasts for 60 minutes. The instrument measures gaseous compounds by sampling air from the chamber at 1.7 L min$^{-1}$ for 30 minutes. At the same time, particles in the chamber are collected onto a polytetrafluoroethylene filter with the sampling rate from 1 to 5 L min$^{-1}$ depending on the aerosol mass concentrations. A gradually heated nitrogen gas flows over the filter, evaporating oxidized organic species and transporting them into the CIMS for detection, with a temperature ramp period of 10 minutes, a soak period of 15 minutes, and a cooling phase of 5 minutes."

(2) We agree with the reviewer that CPS can be influenced by the aerosol mass on the filter. However, all data in Figure 4 come from the same experiment (Exp. 7) and are intended to illustrate the trends in monomer and dimer formation. We have revised the caption of Figure 4 to clarify the data are from Exp. 7: "The time series of major particle-phase products from styrene+NO$_3$ oxidation measured by FIGAERO-CIMS (Exp. 7), including: (a) monomeric styrene ONs; and (b) dimeric styrene ONs." Similarly, we compare the trends observed in FIGAERO-CIMS and HR-ToF-AMS within individual experiments, as shown in Figure S5 and mentioned in lines 248-250: "We also compare the NO$_{3,Org}$ measured by HR-ToF-AMS and speciated ONs measured by FIGAERO-CIMS, which show similar trends (Figure S5)." In addition, since styrene-derived SOA standards are available for mass calibration and we did not perform voltage scanning to quantify the products measured by FIGAERO-CIMS, we have chosen to use CPS instead of quantified units such as ppt or ng m$^{-3}$ to avoid potential misinterpretation.

*6. Line 171-172. The author determined the density by using SMPS and AMS. They found the SOA density is 1.35 g cm-3. However, what is the uncertainty for this value? In this whole experiment, is the SOA density always 1.35 g cm-3? I do not believe it. In addition, the author used this density value to calculate all styrene experiments. The SOA density can be changed by chemical compositions. Therefore, the density should be variable in the series of styrene experiments.*

**Response**

Since we use SMPS and HR-ToF-AMS size distributions to determine the density, the uncertainty mainly arises from particle diameter measurements by the DMA and HR-ToF-AMS. The DMA sizing uncertainty is determined to be 3.5% using polystyrene latex spheres (PSL). Additionally, we conduct size calibration for the HR-ToF-AMS using sized-selected (DMA) ammonium nitrate particles. The uncertainty calculated from the size calibration parameterization is small (0.04%); hence the AMS sizing uncertainty is taken to the be same as the DMA (3.5%).

Although the density can be influenced by chemical composition, the overall composition does not change drastically based on FIGAERO-CIMS measurements when varying the precursor concentration while maintaining a constant styrene-to-oxidant (N$_2$O$_5$) ratio. As shown in Figure R1, the density stays fairly constant over time. We calculate the average density to be $1.35 \pm 0.05$ g cm$^{-3}$.

[Figure]

Figure R1. The time series of density changes in Exp. 10

*7. Line 172-174. The author used the SMPS and aerosol density to calculate the density. How do you separate the SOA and seed particle mass?*

**Response**

We conducted extra experiments without seed particles to determine the density specifically. We mentioned this in lines 108-109: "It is noted that Exp. 9 and 10 do not involve seed particle injection and are conducted specifically to determine the density of styrene+NO₃ SOA".

*8. Line 201-204. What are the decay rates for CxHy, CxHyO, CxHyOzN, CxHyOz, CxHyON? Please provide them.*

**Response**

The decay rates for CxHy, CxHyO, CxHyOz, CxHyON, and CxHyOzN are $1.23\times10^{-3}$ min$^{-1}$, $1.23\times10^{-3}$ min$^{-1}$, $8.82\times10^{-4}$ min$^{-1}$, $5.72\times10^{-4}$ min$^{-1}$, and $1.08\times10^{-3}$ min$^{-1}$, respectively.

*9. Figure 4 shows the time series of major compounds by FIGAERO-CIMS measurements. Which experiments did you plot this figure?*

**Response**

Thanks for pointing this out. The data are from Exp. 7. We added the caption of Figure 4: "The time series of major particle-phase products from styrene+NO₃ oxidation measured by FIGAERO-CIMS (Exp. 7), including: (a) monomeric styrene ONs; and (b) dimeric styrene ONs."

*10. 256-258. FIGAERO-CIMS employing iodide ions for chemical ionization can only measure the highly oxygenated organic compounds. The Sensitivity of Iodide-CIMS to compounds depends on their polarity and hydrogen bonding capability and is strongly influenced by molecular geometry and steric factors (Caldwell et al., 1989). Some organic nitrate with weak polarity and bonding could be completely insensitive by I-CIMS. In this manuscript, the organic nitrate molecular weight is 182.7-184.0. What is uncertainty or error for those values?*

**Response**

We agree with the reviewer. According to Lee et al. (2016, DOI: 10.1073/pnas.1508108113), simple alkyl or keto nitrates are often underestimated or not detected. Additionally, the molecular weight of ONs represents the average molecular weight per nitrate functional group, estimated from FIGAERO-I-CIMS data under the assumption of uniform sensitivity across detected species. As a result, the molecular weight of ONs in this study may be either overestimated or underestimated. However, due to the limitations of the FIGAERO-I-CIMS, which is the only instrument available to us, we have chosen to clarify these limitations rather than introduce an uncertainty estimate. In lines 257-263: "Where $MW_{pON}$ refers to the average molecular weight of $_pON$ estimated from FIGAERTO-CIMS data. Assuming uniform sensitivity among detected species, $MW_{pON}$ is similar across different experiments, within the range of 182.7-184.0 g mol$^{-1}$. $MW_{NO_2,ON}$ is the molecular weight of the nitrogen-containing moiety of ONs (*i.e.*, $NO_2$, 46 g mol$^{-1}$) measured by the HR-ToF-AMS, as discussed in detail in a recent study by Takeuchi et al. (2024). It is noted that given the limitation of FIGAERO-CIMS, which can lead to the underestimation of simple alkyl or keto nitrates (Lee et al., 2016), as well as potential differences in sensitivity among detected species, the $MW_{pON}$ may vary."

*11. Line: 267-268. Why do you think the non-nitrated organic species (C8H8O5) is from the hydrolysis of ONs?*

**Response**

As shown in Figure 5b, we compare the chemical composition differences of SOA under dry and 70% RH conditions using FIGAERO-CIMS spectra. After normalization, we subtract the mass spectrum of SOA from Exp. 12 (RH ~70%) from that of Exp. 2 (RH <3%). A negative signal for a compound indicates that it is generated under RH ~70% conditions. From Figure 5b, we observe that the signal for the non-nitrated organic species ($C_8H_8O_5$) is negative, suggesting its formation under RH ~70%, likely due to the hydrolysis of ONs. In our previous hydrolysis study (Wang et al., 2021, DOI: 10.1021/acs.est.1c05310), the loss of the nitrooxy group under hydrolysis of ONs is found to be a common process. We revise the sentence in lines 273-274: "We also observe the enhancement of non-nitrated organic species (*i.e.*, $C_8H_8O_5$) in the humid experiment, which could be formed from hydrolysis of ONs."

*12. The C8H9NO4 have two peaks in the thermal desorption profile in figure S8. It indicates that it could have isomer on this molecule. However, the author only used the first peak (24.6°C) to calculate the volatility. Please classify it.*

**Response**

The decomposition of dimers in FIGAERO-CIMS can result in an additional peak in the thermal desorption profile. Therefore, we believe the second peak may originate from the thermal decomposition of dimers (Lopez-Hilfiker et al., 2016, DOI: DOI: 10.1021/acs.est.5b04769). We have clarified this point in lines 319-322: "It is noted that although the FIGAERO thermal desorption profile of $C_8H_9NO_4$ shows two peaks, we use the first peak (24.6 °C) to calculate the volatility, as the second peak (around 70 °C) may result from thermal decomposition of dimers (Lopez-Hilfiker et al., 2016; Yang et al., 2021). "

R-5

*13. Line 330-336. The C8H9NO4 and C8H7NO4 from the reference are nitroaromatic compounds. Aromatic nitro compounds are organic molecules that contain a nitro group (-NO2) attached to an aromatic ring. However, nitro group (-NO2) did not attach to the aromatic ring in this manuscript, as shown in Figure 6. Therefore, why do you think styrene hydroxy nitrate (C8H9NO4) and ketone nitrate from styrene-derived ONs accounted for nitroaromatic compounds?*

**Response**

Sorry for the confusion. We have revised the expression in lines 431-435 to improve clarity: "Previous studies have suggested nitrogen-containing products measured in ambient environments are nitroaromatic compounds (R-NO$_2$) formed from aromatic VOCs (Kong et al., 2021; Wang et al., 2021a; Salvador et al., 2021). Our work introduces an alternative perspective, suggesting that ONs could also be nitrogen-containing products from the oxidation of aromatic VOCs in the atmosphere." While C$_8$H$_9$NO$_4$ and C$_8$H$_7$NO$_4$ are classified as nitroaromatic compounds in reference, we propose an alternative interpretation, suggesting that they are styrene-derived ONs instead.

*14. Table 1. The uncertainties in hydrocarbon concentration and aerosol mass loading were 1 standard deviation in this manuscript. However, this is not uncertainty. This is standard deviation. Uncertainty of aerosol mass from SMPS could be from 27%-31% (Buonanno et al., 2009). Please provide the uncertainty of hydrocarbon concentration and aerosol mass loading from your GC-FID and SMPS. And how do you calculate the uncertainty?*

**Response**

We agree with the reviewer to revise the uncertainty of hydrocarbon concentration and aerosol mass loading. For hydrocarbon concentration, the uncertainty is mainly from GC-FID measurement (3.5%). The uncertainty of GC-FID measurement is from GC calibration of styrene. For aerosol mass loading, the uncertainty (6.4%) is from the diameter of particle selected by the DMA (3.5%), counting uncertainty from CPC measurement (4%), and density (3.5%). The uncertainty of DMA (3.5%) is determined from polystyrene latex spheres (PSL) calibrations. The counting uncertainty from CPC measurement is 4% based on our previous work (Takeuchi et al., 2024). In the footnotes of Table 1, we clarify the uncertainty: "[b.] Uncertainties in hydrocarbon concentrations are calculated from 3.5% uncertainty in hydrocarbon concentration measured by GC-FID. [c.] Density is 1.35 ± 0.05 g m$^{-3}$, calculated from the comparison of HR-ToF-AMS and SMPS size distribution data. [d.] Uncertainties in aerosol mass loading are estimated based on uncertainty in aerosol volume concentration measured by the SMPS (5.3%) and uncertainty in SOA density. [e.] These numbers are not reported because the density of SOA in humid experiments is not available. [f.] Uncertainties in SOA mass yields are propagated from the uncertainty associated with hydrocarbon concentration and aerosol mass loading." Table 1, and Figure S7 have been revised accordingly.

**Reviewer: 2**

**General comments**

*This study reports the formation of SOA from NO$_3$ radical oxidation of styrene through a series of environmental chamber experiments that were conducted under varying RH conditions. The SOA yields, chemical composition, and hydrolysis lifetimes of organic nitrates were characterized using a suite of complementary instruments. Overall, this study provides very useful information that can improve the current understanding of styrene SOA formation and hydrolysis of organic nitrates derived from aromatic VOC in aerosol water. I have some questions listed below for the authors' consideration.*

**Response**

We appreciate the viewpoints of the reviewer. Each of the comments has been addressed and detailed in our point-by-point response below.

**Comment**

*1. Lines 79-80: "The authors stated that "To our knowledge, there is no study on hydrolysis of ONs formed from oxidation of aromatic compounds." This statement is not accurate, as there was study on the hydrolysis of organonitrate functional groups in SOA from the oxidation of 1,2,4-trimethylbenzene (TMB) (Liu et al. 2012). The authors also cite this reference in Lines 380-382. Please revise this statement.*

**Response**

Thanks for pointing this out. We revise in lines 77-78: "To our knowledge, there is only one study on hydrolysis of ONs formed from oxidation of aromatic compounds (1,2,4-trimethylbenzene) (Liu et al., 2012). More studies focus on biogenic ONs."

*2. Lines 109-110: Please elaborate on why the initial styrene concentrations of 9.5-155.2 ppb were used in chamber studies. Do they correspond to the known emission inventories, or for any other reasons?*

**Response**

We thank the reviewer for their comment regarding the styrene concentration. The observed ambient concentration of styrene is typically at the ppt level. In typical laboratory chamber experiments, usually higher levels are used to ensure good instrument S/N ratios and formation of detectable amounts of SOA. Considering the detection limit of GC-FID, we decided to use a concentration range of approximately 10–150 ppb. Since the SOA yield curve extends across both low and high concentrations, the reported SOA yield data remain relevant despite the higher experimental concentrations compared to ambient levels.

*3. Lines 219-221: Is it possible to use different analytical methods that do not require thermal desorption (e.g., LC/ESI-MS) to confirm that the presence of monomeric species are true SOA constituents, not a result from thermal decomposition of dimeric species?*

**Response**

That is a great idea. We will attempt to collect filter samples and use LC/ESI-MS or LC/APCI-MS in future work to confirm the presence of monomeric species.

*4. Lines 261-268: Can the authors elaborate on how they quantitatively determine the hydrolysis lifetime to be less than 30 minutes and what parameters may influence this estimate? Does the hydrolysis lifetime refer to $\tau = 1/k$? I did not see any rate constants (k) calculated or discussed in the manuscript.*

**Response**

We thank the reviewer for raising this comment on hydrolysis lifetimes. As illustrated in Figure 5a, the time series of pON/OA stabilizes fairly quickly (< 30 mins). If we utilize pseudo first-order rate equations to assess the hydrolysis lifetimes at 70% RH and 50% RH, the corresponding hydrolysis lifetimes (1/k) are 18.8 ± 1.9 minutes and 29.5 ± 8.7 minutes for 70% RH and 50 % RH, respectively. They are similar within certainty. Nevertheless, we add this information in lines 264-268 to clarify: "As illustrated in Figure 5a, the time series of $_p$ON/OA stabilizes fairly quickly, similar to what we have

observed previously for monoterpene systems (Takeuchi and Ng, 2019). If we utilize pseudo first-order rate equations to assess the hydrolysis lifetimes at 70% RH and 50% RH, the corresponding hydrolysis lifetimes are 18.8 ± 1.9 minutes and 29.5 ± 8.7 minutes for 70% RH and 50 % RH, respectively. Considering the uncertainty, we report the hydrolysis lifetime to be less than 30 minutes. (Takeuchi and Ng, 2019)."

*5. Lines 359-362 and 431-432: Please specify what parameters need to be calculated by density functional theory to support the proposed mechanisms. Given that the proposed mechanisms are based on molecular formulas, not confirmed molecular structures with functional group information, I would suggest clarifying what specific information should be obtained in further experimental studies to validate the proposed formation mechanisms (e.g., confirming the molecular structures and detecting the critical intermediates).*

**Response**

Thank you very much for the suggestion. We revise the sentences in lines 370-372: "Further experimental studies and density functional theory work are necessary to confirm the structures and formation of these dimeric compounds as well as detect the critical intermediates to validate the proposed mechanism." and in lines 441-443: "Further density functional theory calculations and experimental work are needed to provide additional evidence for confirming the molecular structures and identifying critical intermediates to validate our proposed mechanisms."

*6. Lines 856-860 (Figure 5 a): In the time series data of the ratio of particulate ONs (pON) to total organic aerosols (OA) in Exp. 2 (RH<3%), Exp. 11 (RH~50%), and Exp. 12 (RH~70%), the signal for RH~50% drops more rapidly than the signal for RH~70%. Does this imply a faster hydrolysis rate? Can the authors provide some explanations for this observation?*

**Response**

As we mentioned in the response of Q4, the time series of pON/OA stabilizes fairly quickly. If we utilize pseudo first-order rate equations to assess the hydrolysis lifetimes at 70% RH and 50%, the corresponding hydrolysis lifetimes (1/k) are 18.8 ± 1.9 minutes and 29.5 ± 8.7 minutes for 70% RH and 50 % RH, respectively. However, since no sudden, drastic change in pON/OA is observed except for a few initial data points, the reason maybe that the rate of pON hydrolysis may be fast enough such that the decay trend of pON compared to OA is not visibly manifested (Takeuchi and Ng, 2019, DOI: /10.5194/acp-19-12749-2019). Therefore, we conclude that the hydrolysis lifetime of hydrolyzable pON for styrene-derived ON shall be less than 30 minutes. We add the sentence in lines 264-268 to avoid the misunderstanding: "As illustrated in Figure 5a, the time series of $_p$ON/OA stabilizes fairly quickly, similar to what we have observed previously for monoterpene systems (Takeuchi and Ng, 2019). If we utilize pseudo first-order rate equations to assess the hydrolysis lifetimes at 70% RH and 50% RH, the corresponding hydrolysis lifetimes are 18.8 ± 1.9 minutes and 29.5 ± 8.7 minutes for 70% RH and 50 % RH, respectively. Considering the uncertainty, we report the hydrolysis lifetime to be less than 30 minutes. (Takeuchi and Ng, 2019)."

*7. Lines 862-869 (Figure 6): Can the authors label or clarify which products were measured and confirmed in the current study and which were proposed that need to be validated in further studies?*

**Response**

Thank you very much for this suggestion. We revised the caption of Figure 6: "All chemical structures in the formation mechanisms are proposed based on the molecular formulas of the detected species (shown beneath the structures) or fragments measured by HR-ToF-AMS and require validation through further experiments or theoretical calculations."

---

## Author Response (AR2)

**Point-by-point response to reviewer comments**

Manuscript ID:  egusphere-2024-3849
Title: "Secondary Organic Aerosol Formation from Nitrate Radical Oxidation of Styrene: Aerosol Yields, Chemical Composition, and Hydrolysis of Organic Nitrates"
Author(s): Wang, Yuchen; Xiang Zhang; Yuanlong Huang; Yutong Liang; Nga L. Ng

We thank the editor for the thoughtful comments. Each of the comments has been addressed and detailed in our point-by-point response below. The exact comment from the editor is in black and italic style while our response is in blue and normal format in this document. The revised text as it appears in the revised manuscript is in green.

**Comment**

*1. Did you consider a potential bias of sulfate mass concentrations from AMS measurements due to coating of the seed particles and corresponding change in the collection efficiency? How could this affect your analysis?*

**Response**

According to previous work by Bahreini et al. (2005, DOI: 10.1021/es048061a), the sulfate signal in HR-ToF-AMS increased following SOA formation, indicating improved collection efficiency (CE) of organic-coated seed particles. However, in this work, the SOA yields are calculated from the SMPS data. We did not calculate collection efficiency (CE) as we did not need to use this value in our analysis.

*2. Figure 3a: Please add the relative abundance of monomeric and dimeric products (or CHO, CHON, CHON2) as a function of initial styrene concentrations and discuss the result in the context of typical atmospheric styrene concentrations.*

**Response**

We added a new figure (Figure S4) to show the relative percentage contributions of CHO, CHON, $CHON_2$ as well as dimers and monomers. We also added the following to the main text in lines 216-218: "The relative percentage contributions of CHO, CHON, $CHON_2$, as well as dimers and monomers, remained relatively stable across various experiments with differing initial styrene concentrations (Figure S4)." We also mention in lines 435-438: "As the fractions of monomers dimers are relatively constant in experiments spanning a wide range of styrene concentrations, this may suggest that dimeric ONs are also important products when $NO_3$ reacts with typical atmospheric levels of styrene."

*3. Figure 5a: Explain why the hydrolysis lifetimes are: 18.8$\pm$1.9 min at 70% RH (Shorter lifetime despite slower decay rate?) and 29.5$\pm$8.7 min at 50% RH (Longer lifetime despite faster decay rate?) although Figure 5a shows a faster decay of the pON/OA ratio at 50% RH compared to 70% RH?*

**Response**

Since no sudden, drastic change in pON/OA was observed except for a few initial data points (Figure R1), the rate of pON hydrolysis may be fast enough such that the decay trend of pON compared to OA is not visibly manifested, as we mentioned in our previous hydrolysis paper (Takeuchi and Ng, 2019, DOI: /10.5194/acp-19-12749-2019). To avoid misunderstanding, we revised the sentence in lines 266-272: "As illustrated in Figure 5a, the time series of $_pON/OA$

stabilizes fairly quickly, similar to what we have observed previously for monoterpene systems (Takeuchi and Ng, 2019). If we utilize pseudo first-order rate equations to assess the hydrolysis lifetimes at 70% RH and 50% RH, the corresponding hydrolysis lifetimes are $18.8 \pm 1.9$ minutes and $29.5 \pm 8.7$ minutes for 70% RH and 50 % RH, respectively. Considering that a drastic change in the pON/OA ratio is not observed except for a few initial data points, the rate of pON hydrolysis may be fast enough that the decay trend of pON relative to OA is not visibly apparent. Therefore, we report the hydrolysis lifetime to be less than 30 minutes (Takeuchi and Ng, 2019)."

[Figure]

Figure R1. Time series data of the ratio of particulate ONs ($_p$ON) to total organic aerosols (OA) in Exp. 2 (RH<3%), Exp. 11 (RH~50%), and Exp. 12 (RH~70%) in first 20 minutes.

*4. Figure S6: Consider changing the organic mass axis (e.g. log scale) to make the atmospheric relevant range better visible and discuss a potential extrapolation of your results to this range. Indicate how the lines given are calculated also in the caption.*

**Response**

Thanks for the suggestion. We have added a new panel with log scale to Figure S6 (now Figure S7 in the latest version). As seen in the figure, the organic mass concentrations in our experiments (covering up to highly polluted areas) are much lower than those used in prior studies. We have revised the figure caption: "Comparison of SOA yields of styrene oxidation systems in this work and in literature (Yu et al., 2022; Tajuelo et al., 2019; Ma et al., 2018; Na et al., 2006; Schueneman et al., 2024) (a) on a linear scale and (b) on a logarithmic scale. The SOA yields and $\Delta M_O$ in Schueneman et al., (2024) and in Tajuelo et al., (2019) are extracted by WebPlotDigitizer. The higher end of the $\Delta M_O$ range in this study corresponds to highly polluted environments. The lines are SOA yield curves obtained by fitting the yield data to the Odum two-product model (Odum et al., 1996, 1997): $Y = \Delta M_O \left[ \frac{\alpha_1 K_1}{1 + K_1 M_O} + \frac{\alpha_2 K_2}{1 + K_2 M_O} \right]$, with coefficients either taken directly from the published papers ( Tajuelo et al., 2019; Na et al., 2006) or determined by ourselves using the published yield data points (Yu et al., 2022; Ma et al., 2018; Schueneman et al., 2024)."